Original research

# Intersection of reproductive coercion and intimate partner violence: cross-sectional influences on women's contraceptive use in Burkina Faso, Côte d'Ivoire and Kenya

Shannon N Wood ,[1,2] Haley L Thomas,[1] Mary Thiongo,[3] Georges Guiella,[4] Fiacre Bazié,[4] Yentéma Onadja,[4] Rosine Mosso,[5] Raimi Fassassi,[5] Peter Gichangi,[3,6,7] Michele R Decker[1,2,8]

For numbered affiliations see end of article.

**Correspondence to**
Dr Shannon N Wood;
swood@jhu.edu

## ABSTRACT

**Objectives** Among nationally representative cross-sections of women in need of contraception from Burkina Faso, Côte d'Ivoire and Kenya, we aimed to: (1) examine the intersection of past-year physical/sexual intimate partner violence (IPV), emotional IPV and reproductive coercion (RC) and (2) assess the impact of physical/sexual IPV, emotional IPV and RC on women's contraceptive use outcomes, including current contraceptive use, method type and covert use.

**Design** The present analysis uses cross-sectional female data collected in Burkina Faso (December 2020–March 2021), Côte d'Ivoire (October–November 2021) and Kenya (November–December 2020).

**Settings** Burkina Faso, Côte d'Ivoire and Kenya

**Participants** Analytical samples were restricted to partnered women with contraceptive need who completed a violence module (Burkina Faso n=1863; Côte d'Ivoire n=1105; Kenya n=3390).

**Primary and secondary outcome measures** The exposures of interest—past-year emotional IPV, physical/sexual IPV and RC—were assessed using abridged versions of the Revised Conflict and Tactics Scale-2 and Reproductive Coercion Scale, respectively. Outcomes of interest included current contraceptive use, contraceptive method type (female controlled vs male compliant), and covert contraceptive use, and used standard assessments.

**Results** Across sites, 6.4% (Côte d'Ivoire) to 7.8% (Kenya) of women in need of contraception experienced RC; approximately one-third to one-half of women experiencing RC reported no other violence forms (31.7% in Burkina Faso to 45.8% in Côte d'Ivoire), whereas physical/sexual IPV largely occurred with emotional IPV. In multivariable models, RC was consistently associated with covert use (Burkina Faso: aOR 2.84 (95% CI 1.21 to 6.64); Côte d'Ivoire: aOR 4.45 (95% CI 1.76 to 11.25); Kenya: aOR 5.77 (95% CI 3.51 to 9.46)). Some IPV in some settings was also associated with covert use (emotional IPV, Burkina Faso: aOR 2.99 (95% CI 1.56 to 5.74); physical/sexual, Kenya: aOR 2.35 (95% CI 1.33 to 4.17)).

**Conclusions** Across settings, covert use is a critical strategy for women experiencing RC. Country policies

### STRENGTHS AND LIMITATIONS OF THIS STUDY

⇒ Concurrent measurement of reproductive coercion (RC) and intimate partner violence within three national samples (Burkina Faso, Cote d'Ivoire and Kenya) afforded examination of concurrent violence experiences.

⇒ Given inclusion of comprehensive measures, there was a unique opportunity to examine the interplay of violence experiences and their impact on three contraceptive use outcomes (current contraceptive use, contraceptive method type, covert use).

⇒ Findings are cross-sectional, limiting conclusions surrounding temporality of associations.

⇒ While RC measurement uses the pregnancy coercion subscale from the RC Scale, the most comprehensive measure for RC to date, RC items do not account for a wide range of RC behaviours, and therefore, may underestimate the true burden of RC in these contexts.

must recognise RC as a unique form of violence with profound implications for women's reproductive health.

## INTRODUCTION

Intimate partner violence (IPV) is pervasive globally—recent evidence highlights that 27% of women are affected by physical or sexual IPV.[1] Partners can further harm women's health and livelihoods through psychological and economic abuse and coercive control.[2,3] Reproductive coercion (RC), or partner control over contraceptive use and reproductive health trajectories,[4,5] was initially conceptualised as a unique form of violence that could occur in tandem with physical or sexual IPV.[6,7] RC,[6,7] like IPV,[8] has a negative impact on women's reproductive health trajectories and has been linked to

decreased use of effective contraception and increased risk of unintended pregnancy.

Current evidence examining the reproductive health impact of these leading forms of violence has been concentrated within high-income contexts, namely, the USA[6 7] and Australia.[9] Recent literature, however, highlights that RC is also common in sub-Saharan Africa, though behaviours, intentions and motivations may differ in comparison to high-income contexts.[10–13] Specifically, partners may enact a broader range of RC behaviours with the means of asserting power over reproductive decision-making, including seeking to prevent pregnancy against a woman's wishes via behaviours such as forced contraceptive use.[10]

Understanding the impact of abusive partner dynamics on women's reproductive health in sub-Saharan Africa is critical given that sub-Saharan African women experience disproportionate adverse health burdens; specifically, they have the highest global rates of unintended pregnancy,[14] maternal mortality[15] and unsafe abortion.[16] Multicountry studies have sought to understand the impact of IPV on contraceptive use with inconclusive findings.[17] Studies have reported increased contraceptive and covert use among IPV survivors,[12] whereas others have indicated decreased contraceptive use.[18] Moreover, such studies have largely failed to disentangle the impact of RC and emotional abuse (ie, manipulation, intimidation, verbal abuse) on women's reproductive health outcomes, though women may be pressured or shamed when seeking to delay or limit childbearing; as such, these forms of violence may significantly impact contraceptive use and continuation.[11 13 17]

Inclusion of comprehensive RC measures within recent population-based surveys allows for the examination of the interplay of violence experiences and their impact on contraceptive use outcomes. To date, analyses examining linkages between unique forms of violence and contraceptive use have been limited given lack of RC and IPV measures concurrently included in national surveys. In three national samples across East and West Africa, this study aimed to: (1) examine the intersection of physical/sexual IPV, emotional IPV and RC and (2) assess the impact of IPV and RC on women's contraceptive use outcomes (current contraceptive use, method type and covert use).

## METHODS
### Study contexts
In 2019, Burkina Faso, Côte d'Ivoire and Kenya ranked similarly on the United Nations Development Programme Gender Inequality Index (Burkina Faso 0.594, rank 147 out of 162[19]; Côte d'Ivoire 0.638, rank 153[20]; Kenya 0.518, rank 126).[19] All three countries have ratified the Convention on the Elimination of all Forms of Discrimination Against Women and have taken steps to protect women from gender-based violence (GBV),[21] however, RC has

not been formally included as form of violence within national policies.

### Overview of Performance Monitoring for Action and GBV module
Analyses use data from Performance Monitoring for Action (PMA), which administers annual population-based panel and cross-sectional questionnaires at the household, female and facility levels using a multistage cluster design allowing for nationally and regionally representative estimates. Additional details are available at pmadata.org.

This analysis uses cross-sectional female data collected in Burkina Faso (December 2020–March 2021), Côte d'Ivoire (October–November 2021) and Kenya (November–December 2020). Sites were selected for the present analysis due to implementation of the PMA GBV module, which allowed for examination of both IPV and RC experiences.

All female household members ages 15–49 were eligible for the PMA female survey. In accordance with ethical best practices,[22 23] only one female per household was eligible for the GBV module. If there was more than one eligible female per household, one woman per household was randomly selected.

### Training and ethical protections
Data were collected by trained resident enumerators (REs) using mobile phones equipped with Open Data Kit software; privacy checks were built into the survey. Prior to implementation, all REs completed a GBV-specific training, which emphasised maintaining confidentiality and privacy, asking non-judgemental questions, monitoring for emotional upset (ie, crying, agitation, restlessness) and referring women to support services. In line with best practices,[22 23] all participants, regardless of violence disclosure or selection into the module, received a referral sheet for GBV and health resources.

### Patient and public involvement
The GBV module was included within select PMA surveys at the request of the Ministries of Health. Results are disseminated to in-country stakeholders at national and local events.

### Analytical samples
One woman per household was randomly selected to complete the GBV module (n=4289 Burkina Faso; n=2752 Côte d'Ivoire; n=6870 Kenya); participants did not complete the GBV module if there were privacy concerns (n=60 Burkina Faso; n=88 Côte d'Ivoire; n=119 Kenya). IPV and RC items were only administered to married or partnered women for a final sample of 3048 women in Burkina Faso, 1852 women in Côte d'Ivoire and 4355 women in Kenya.

The analytical sample was further restricted to women in need of contraception, defined as being sexually active in the last 12 months, wanting to wait more than 12 months to have another child or not wanting any more

children, not currently pregnant and fecund, for a final sample of n=1863 Burkina Faso, n=1105 Côte d'Ivoire, n=3390 Kenya. Analytical samples float to accommodate small amounts of missing data (<1%).

## Measures

Three independent variables classified past-year experiences of violence perpetrated by a husband/partner (emotional IPV, physical/sexual IPV and RC) via behavioural assessment. Past-year IPV was measured via five-item abbreviated version of the Revised Conflict and Tactics Scale-2.[24] Specifically, emotional IPV was assessed via affirmative response to a single behaviour (In the past 12 months has your husband/partner insulted you, yelled at you, screamed or made humiliating remarks). Physical/sexual IPV was assessed via affirmative response to any of the following items: in the past 12 months has your husband/partner slapped, hit, or physically hurt you; threatened with a weapon or attempted to strangle or kill you; pressured or insisted on having sex when you did not want to (without physical force); physically forced you to have sex when you did not want to. Past-year RC was measured via six items from the pregnancy coercion subscale of the RC scale[25] and previously adapted to the sub-Saharan African context,[12] and included the following items: in the past 12 months has your husband/partner mistreated you for wanting to use family planning; forced or pressured you to become pregnant; said he would leave if you did not get pregnant; said he would have a baby with someone else if you did not get pregnant; took away your family planning or prevented you from going to the clinic for family planning; hurt you physically because you did not get pregnant. The condom-manipulation subscale of the RC Scale was not included given that condoms are more likely to be used in these settings for preventing HIV/STIs than pregnancy.

Three dichotomous dependent contraceptive use variables (current contraceptive use, method classification (female-controlled or male compliant) and covert contraceptive use) were examined using standard assessments.[26] Current contraceptive use was determined from a single no/yes survey item. Method mix classification (binary-male compliant/female controlled) was generated based on level of partner involvement needed in the reported contraceptive method. Specifically, current contraceptive users were asked to identify all contraceptive methods being used; the respondent's most effective method was then categorised as either a male compliant method or a female-controlled method, with female-controlled methods being those that women could use without partner involvement (female sterilisation, implant, intra-uterine device (IUD), injectables, pills, emergency contraception (EC), lactational amenorrhoea method (LAM)) and male-compliant methods being those requiring men to actively use or accept women's use, including cycle tracking methods given necessitation of compliance in timing of sexual activity (male sterilisation, male condom, female condom, diaphragm, foam/jelly, standard days/

cycle beads, rhythm, withdrawal, other traditional). Direct covert use assessment was used to determine covert contraceptive use among current users,[27] where each contraceptive user was asked the following question: 'does your husband/partner know that you are using method reported?' (response categories: no/yes).

Several sociodemographic and relationship characteristics were explored as potential confounders between violence and contraceptive use outcomes using standard assessments,[26] including residence (urban/rural); parity (0, 1–2, 3+), marital status (married/living with a partner), polygyny (yes/no) and employment (yes/no).

## Statistical analysis

All analyses were stratified by site. Sociodemographic, relationship, and sexual and reproductive health characteristics were examined overall and among those with contraceptive need. The prevalence of past-year RC was calculated overall and by item; prevalence of past-year IPV was calculated overall and by IPV subtype (emotional, physical/sexual). Overlapping experiences of emotional IPV, physical/sexual IPV and RC were examined via Venn diagrams. Bivariate and multivariable logistic regression models were used to separately examine each contraceptive outcome by experience of violence; specifically, three models were used: (1) bivariate, (2) adjustment for sociodemographic and characteristics only and (3) adjustment for sociodemographic and relationship characteristics and other types of violence. Sociodemographic and relationship characteristics were examined for multicollinearity, and ultimately, the third model was retained given <10% difference in effect estimates between models. All analyses were conducted in Stata V.16 and weighted to account for survey design.

## RESULTS

In Burkina Faso and Kenya, most women lived within rural areas, whereas in Côte d'Ivoire, residence was split between urban and rural localities (table 1). Approximately, 90% of partnered Burkinabe and Kenyan women were married compared with large proportions of women from Côte d'Ivoire lived with a partner outside of marriage (39.4%). Polygynous unions were highest in Burkina Faso (28.2%) and lowest in Kenya (11.4%). More than three-quarters of women in need of contraception were currently using contraception in Kenya (77.6%), whereas this proportion was substantially lower in Burkina Faso (52.6%) and Côte d'Ivoire (49.2%). In Kenya and Côte d'Ivoire, short-acting hormonal methods were the most used contraceptive methods (48.4% and 49.9%, respectively); in Burkina Faso, higher proportions of women used long-acting reversible contraceptives (45.5%). Approximately, 11% of contraceptive users in Burkina Faso and Côte d'Ivoire used covertly; this proportion was slightly lower in Kenya (7.6%).

Across sites, emotional IPV was the most prevalent type of violence experienced among women in need of

**Table 1** Demographic characteristics of married/partnered women selected for GBV module, by country

| | Burkina Faso | | Côte d'Ivoire | | Kenya | |
|---|---|---|---|---|---|---|
| | Married/partnered women (n=3048) | FP need (n=1863) | Married/partnered women (n=1852) | FP¶ need (n=1105) | Married/partnered women (n=4355) | FP need (n=3390) |
| | %* | | | | | |
| **Sociodemographic** | | | | | | |
| Residence | | | | | | |
| Urban | 19.6 | 20.2 | 48.5 | 47.7 | 30.6 | 30.8 |
| Rural | 80.4 | 79.9 | 51.5 | 52.3 | 69.4 | 69.2 |
| Household wealth tertile | | | | | | |
| Lowest | 35.1 | 37.4 | 39.1 | 39.6 | 37.0 | 36.7 |
| Middle | 33.8 | 32.5 | 31.0 | 31.3 | 33.6 | 33.8 |
| Highest | 31.1 | 30.1 | 29.9 | 29.1 | 29.5 | 29.5 |
| Household composition | | | | | | |
| Does not live with extended family | 63.7 | 64.0 | 57.7 | 59.1 | 76.1 | 76.3 |
| Lives with extended family | 36.3 | 36.0 | 42.3 | 41.0 | 23.9 | 23.7 |
| Age | | | | | | |
| 15–19 | 7.5 | 7.6 | 5.3 | 5.5 | 2.1 | 1.8 |
| 20–29 | 42.0 | 43.3 | 36.1 | 38.8 | 38.7 | 37.9 |
| 30–39 | 34.0 | 32.9 | 40.9 | 41.6 | 39.0 | 40.6 |
| 40–49 | 16.4 | 16.2 | 17.8 | 14.1 | 20.3 | 19.7 |
| Education | | | | | | |
| None | 67.0 | 67.6 | 52.6 | 49.5 | 4.8 | 4.1 |
| Primary | 18.5 | 18.2 | 26.6 | 27.1 | 51.4 | 52.0 |
| Secondary or higher | 14.5 | 14.2 | 20.7 | 23.4 | 43.9 | 43.9 |
| Parity | | | | | | |
| 0 | 4.7 | 0.8 | 6.4 | 1.6 | 3.7 | 1.4 |
| 1–2 | 33.2 | 33.8 | 35.3 | 33.3 | 37.9 | 36.8 |
| 3+ | 62.2 | 65.4 | 58.3 | 65.2 | 58.5 | 61.8 |
| **Relationship dyad** | | | | | | |
| Marital status | | | | | | |
| Married | 90.9 | 92.4 | 64.2 | 60.6 | 89.5 | 89.6 |
| Living with partner | 9.1 | 7.6 | 35.9 | 39.4 | 10.5 | 10.4 |
| Polygamous union | 30.5 | 28.2 | 18.4 | 16.6 | 12.1 | 11.4 |
| Age at marriage | | | | | | |
| ≥15 | 5.3 | 4.9 | 11.9 | 11.8 | 7.0 | 7.1 |
| >15 and <18 | 47.0 | 46.9 | 29.1 | 31.0 | 22.6 | 22.0 |
| ≥18 | 47.8 | 48.2 | 59.0 | 57.2 | 70.4 | 70.9 |
| Partner education | | | | | | |
| None | 62.5 | 63.4 | 40.3 | 39.1 | 3.9 | 3.4 |
| Primary | 21.2 | 21.0 | 24.0 | 22.7 | 44.7 | 45.6 |
| Secondary or higher | 16.3 | 15.6 | 35.7 | 38.2 | 51.4 | 51.0 |
| **Finance** | | | | | | |
| Employed | 29.8 | 30.3 | 47.1 | 47.8 | 48.0 | 49.3 |
| Has savings | 15.3 | 15.4 | 8.7 | 8.7 | 48.0 | 48.1 |
| Level of financial knowledge | | | | | | |
| Not at all or not very | 92.9 | 92.8 | 63.7 | 63.7 | 21.8 | 21.7 |

**Table 1** Continued

| | Burkina Faso | | Côte d'Ivoire | | Kenya | |
|---|---|---|---|---|---|---|
| | Married/partnered women (n=3048) | FP need (n=1863) | Married/partnered women (n=1852) | FP¶ need (n=1105) | Married/partnered women (n=4355) | FP need (n=3390) |
| Somewhat or very | 7.1 | 7.2 | 36.3 | 36.3 | 78.2 | 78.3 |
| Economically reliant on partner for basic needs | 51.8 | 52.4 | 56.2 | 56.4 | 61.9 | 61.4 |
| **SRH** | | | | | | |
| Currently pregnant | 10.4 | -- | 10.3 | -- | 8.1 | -- |
| Feeling when found out about pregnancy* | | | | | | |
| Very/sort of happy | 74.5 | -- | 67.2 | -- | 66.9 | -- |
| Mixed happy and unhappy | 1.8 | -- | 7.3 | -- | 12.9 | -- |
| Sort of/very unhappy | 23.7 | -- | 25.6 | -- | 20.2 | -- |
| Currently using any contraceptives | 36.0 | 52.6 | 32.8 | 49.2 | 66.1 | 77.6 |
| Method mix, among users† | | | | | | |
| Male Compliant | 18.5 | 18.6 | 30.5 | 29.8 | 10.6 | 10.7 |
| Female-controlled (LARC)‡ | 45.7 | 45.5 | 19.8 | 19.5 | 41.5 | 39.4 |
| Female-controlled (short acting) | 35.8 | 35.9 | 49.7 | 50.7 | 47.9 | 50.0 |
| Covert contraceptive use | 12.5 | 11.1 | 12.0 | 11.3 | 7.9 | 7.6 |
| Feeling if you got pregnant now§ | | | | | | |
| Very/sort of happy | 44.3 | 33.7 | 50.4 | 34.8 | 37.3 | 32.5 |
| Mixed happy and unhappy | 4.9 | 5.9 | 8.2 | 10.8 | 16.0 | 17.9 |
| Sort of/very unhappy | 50.8 | 60.4 | 41.4 | 54.4 | 46.6 | 49.7 |

*Among women who are currently pregnant.
†Male compliant (male sterilisation, male and female condoms, diaphragm, foam/jelly, standard days/cycle beads, rhythm, withdrawal, other traditional), female-controlled LARC (female sterilisation, implant, IUD and female-controlled short-acting (injectables, pills, EC, LAM).
‡Includes female sterilisation (n=4 in Burkina Faso, n=0 in Côte d'Ivoire, n=102 in Kenya among the married/partnered samples).
§Among women who are not currently pregnant.
¶FP: Family Planning
**SRH: Sexual and Reproductive Health
EC, emergency contraception; FP, family planning; GBV, gender-based violence; IUD, intrauterine device; LAM, lactational amenorrhoea method; LARC, long-acting reversible contraceptive; SRH, sexual and reproductive health.

contraception (20.1% Kenya, 23.9% Burkina Faso, 29.0% Côte d'Ivoire), followed by physical/sexual IPV (9.9% Burkina Faso, 13.2% Kenya, 14.5% Côte d'Ivoire) and RC (6.4% Côte d'Ivoire, 7.0% Burkina Faso, 7.8% Kenya; table 2). Mistreatment for wanting to use family planning was the most common RC behaviour (5.2% Burkina Faso, 4.4% Côte d'Ivoire and 5.3% Kenya), whereas physical violence as a result of not getting pregnant was rarer (0.3% Burkina Faso, 0.4% Côte d'Ivoire, 1.2% Kenya).

Women in need of contraception experienced multiple types of violence simultaneously, with 10.1%, 12.9% and 12.7% experiencing 2 of more types of violence in Burkina Faso, Côte d'Ivoire and Kenya, respectively (figure 1). When exclusively examining reports of RC (n=139 Burkina Faso; n=72 Côte d'Ivoire; n=251 Kenya), approximately one-third to one-half of women reported RC experience in isolation (31.7% (44/139) Burkina Faso, 45.8% (33/72) Côte d'Ivoire, 35.1% (88/251) Kenya). In Burkina Faso, substantial portions women experiencing RC reported RC in tandem with emotional IPV (36.0% (50/139)), whereas in Côte d'Ivoire and Kenya, approximately one-third reported overlap with both emotional IPV and physical/sexual IPV (36.1% (26/72) and 31.9% (80/251), respectively).

In bivariate analysis, RC was associated with covert contraceptive use in all three sites (p<0.001; table 3). Emotional IPV was associated with increased use of contraception (50.9% no $_{emotional IPV}$ vs 58.2% $_{emotional IPV}$; p<0.05) in Burkina Faso only and increased covert contraceptive use in both Burkina Faso (7.5% no $_{emotional IPV}$ vs 20.9% $_{emotional}$ IPV; p<0.001) and Kenya (6.1% no $_{emotional IPV}$ vs 13.2% $_{emotional}$ IPV; p<0.001). Physical/sexual IPV was associated with covert contraceptive use in Kenya only (6.1% no $_{physical/sexual IPV}$ vs 17.5% $_{physical/sexual IPV}$; p<0.001).

In multivariable models, RC remained consistently associated with increased covert use of contraception across sites (Burkina Faso: adjusted odds ratio (aOR) 2.84; 95% CI 1.21 to 6.64; Côte d'Ivoire: aOR 4.45; 95%

**Table 2** Prevalence of past-year reproductive coercion and IPV among married/partnered women with need for contraception, by country

| | Burkina Faso | Côte d'Ivoire | Kenya |
|---|---|---|---|
| | **% weighted** | | |
| Individual RC items | | | |
| In the past 12 months has your husband/partner… | | | |
| Mistreated you for wanting to use family planning | 5.2 | 4.4 | 5.3 |
| Forced or pressured you to become pregnant | 2.2 | 3.2 | 4.3 |
| Said he would leave if you did not get pregnant | 1.3 | 1.6 | 1.9 |
| Said he would have baby with someone else if you did not get pregnant | 1.3 | 1.9 | 1.9 |
| Took away your family planning or prevented you from going to clinic for family planning | 1.6 | 0.9 | 2.0 |
| Hurt you physically because you did not get pregnant | 0.3 | 0.4 | 1.2 |
| Any experience of RC | 7.0 | 6.4 | 7.8 |
| IPV | | | |
| Emotional IPV | 23.9 | 29.0 | 20.1 |
| Physical/sexual IPV | 9.9 | 14.5 | 13.2 |
| Any experience of IPV | 26.9 | 32.1 | 23.1 |

IPV, intimate partner violence; RC, reproductive coercion .

CI 1.76 to 11.25; Kenya: aOR 5.77; 95% CI 3.51 to 9.46; table 4). Emotional IPV was associated further with covert contraceptive use in Burkina Faso (aOR 2.99; 95% CI 1.56 to 5.74) and physical/sexual IPV with covert contraceptive use in Kenya (aOR 2.35; 95% CI 1.33 to 4.17). In Kenya, emotional IPV was associated with increased odds of current contraceptive use (aOR 1.44; 95% CI 1.04 to 2.01), and in Burkina Faso, emotional IPV was marginally associated with current contraceptive use below the significance threshold (aOR 1.36; 95% CI 0.97 to 1.92; p=0.08). Also, in Burkina Faso, RC experience was associated with non-significant decreases in current contraceptive use (aOR 0.66; 95% CI 0.43 to 1.01; p=0.06). None of the violence forms displayed significant associations with method type.

## DISCUSSION

Leveraging large-scale, population-based samples, this analysis illuminates the prevalence of RC and overlap with physical/sexual and emotional IPV, and its impact on current and covert contraceptive use in Burkina Faso, Côte d'Ivoire and Kenya. Data illustrate that for women experiencing RC who are in need of contraception, RC often occurred in isolation (31.7% in Burkina Faso to 45.8% in Côte d'Ivoire) or coupled with emotional IPV (16.7% in Côte d'Ivoire to 36.0% in Burkina Faso). Accordingly, only situating RC interventions within IPV programmes and response services may overlook the needs of women who are experiencing this unique, yet detrimental, form of violence. Further, screening for physical or sexual IPV alone will fail to capture most RC experiences.

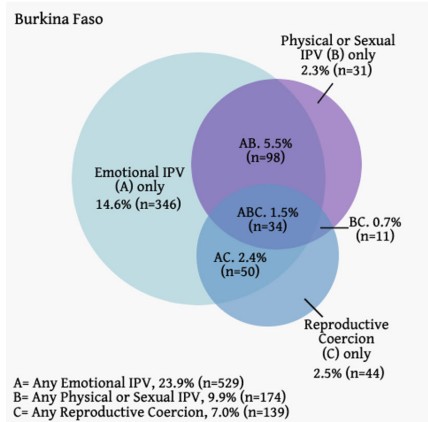
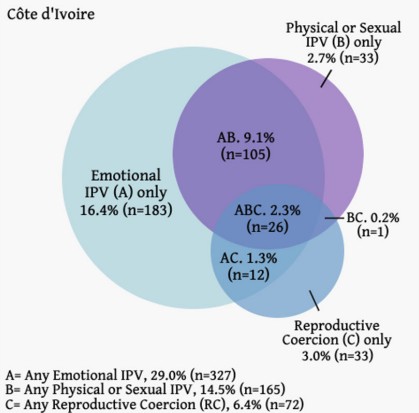
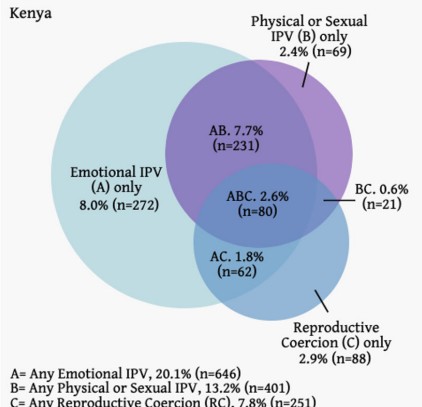

**Figure 1** Venn diagrams of IPV and reproductive coercion experiences, among partnered women with need for contraception, by country. IPV, intimate partner violence.

**Table 3** Bivariate analysis between past-year RC and IPV and contraceptive use outcomes among married/partnered women in need of contraception, per country

| | Burkina Faso | | | | | |
| --- | --- | --- | --- | --- | --- | --- |
| | RC | | Emotional IPV | | Physical/sexual IPV | |
| | No RC | RC | No IPV | IPV | No IPV | IPV |
| | (n=1719) | (n=139) | (n=1332) | (n=529) | (n=1686) | (n=174) |
| | % | | % | | % | |
| Current contraceptive use | 53.2 | 46.2 | **50.9** | **58.2*** | 52.6 | 52.7 |
| Method mix, among users | | | | | | |
| Male compliant | 18.5 | 19.5 | 19.5 | 15.9 | 19.0 | 15.0 |
| Female controlled | 81.5 | 80.6 | 80.5 | 84.1 | 81.1 | 85.0 |
| Covert contraceptive use | **9.9** | **29.3*** | **7.5** | **20.9*** | 10.3 | 17.9 |
| | Cote d'Ivoire | | | | | |
| | RC | | Emotional IPV | | Physical/sexual IPV | |
| | No RC | RC | No IPV | IPV | No IPV | IPV |
| | (n=1026) | (n=72) | (n=778) | (n=327) | (n=940) | (n=165) |
| | % | | % | | % | |
| Current contraceptive use | 49.1 | 54.2 | 48.1 | 52.0 | 48.2 | 54.9 |
| Method mix, among users | | | | | | |
| Male compliant | 30.7 | 19.3 | 31.4 | 26.3 | 30.8 | 24.5 |
| Female controlled | 69.3 | 80.7 | 68.7 | 73.7 | 69.2 | 75.5 |
| Covert contraceptive use | **9.5** | **35.0*** | 9.4 | 15.5 | 10.9 | 13.0 |
| | Kenya | | | | | |
| | RC | | Emotional IPV | | Physical/sexual IPV | |
| | No RC | RC | No IPV | IPV | No IPV | IPV |
| | (n=3135) | (n=251) | (n=2741) | (n=646) | (n=2986) | (n=401) |
| | % | | % | | % | |
| Current contraceptive use | 77.9 | 74.2 | 77.2 | 79.2 | 77.8 | 76.3 |
| Method mix, among users | | | | | | |
| Male compliant | 10.7 | 12.1 | 10.3 | 12.6 | 10.6 | 11.5 |
| Female controlled | 89.4 | 87.9 | 89.7 | 87.4 | 89.4 | 88.5 |
| Covert contraceptive use | **5.6** | **30.8*** | **6.1** | **13.2*** | **6.1** | **17.5*** |

P value from bivariate logistic regression *p<0.05, **p<0.01, ***p<0.001. p<0.05, **p<0.01, ***p<0.001
Bolding represents significance p<0.05.
Male compliant: Male sterilisation, male condom, female condom, diaphragm, foam/jelly, standard days/cycle beads, rhythm, withdrawal, other traditional; female-controlled: female sterilisation, implant, IUD, injectables, pills, EC, LAM.
EC, emergency contraception; IPV, intimate partner violence; IUD, intrauterine device; LAM, lactational amenorrhea method; RC, reproductive coercion.

Notably, experience of past-year RC was consistently associated with increased covert use of contraception across sites (Burkina Faso: aOR 2.84; 95% CI 1.21 to 6.64; Côte d'Ivoire: aOR 4.45; 95% CI 1.76 to 11.25; Kenya: aOR=5.77; 95% CI 3.51 to 9.46), however, similar results were not observed for female-controlled method use. Covert use may prove a promising, yet risky safety strategy for women seeking to avert pregnancy and also experiencing RC.[12] Qualitative data with IPV and RC survivors in Nairobi indicated the cyclic covert use experiences of women—first electing to use contraception covertly given RC, then IPV or RC on partner finding out about use, followed by a subsequent attempt to use a different contraceptive method covertly.[28] While providers may recommend female-controlled methods in light of partner disapproval, such as IUDs or implants, control of method use alone may not be enough to protect against unintended pregnancy when faced with RC. Instead, providers must be aware of RC and inquire about covert use preferences and potential RC experiences during contraceptive counselling to ensure method continuity and aversion of pregnancy without detriment to women's safety. Specifically, understanding previous experiences with side effects will be essential to ensure that women are able to use methods covertly. Further follow-up care will be necessary for women

**Table 4** Multivariable logistic regression between past-rear RC and IPV and contraceptive use outcomes among married/partnered women in need of contraception, per country

| | Burkina Faso | | |
| | RC | Emotional IPV | Physical/sexual IPV |
| | aOR† (95% CI) | | |
|---|---|---|---|
| Current contraceptive use | 0.66 (0.43 to 1.01)± | 1.36 (0.97 to 1.92)± | 0.91 (0.54 to 1.54) |
| Method mix, among users | | | |
| Male compliant | ref | ref | ref |
| Female controlled | 0.81 (0.36 to 1.86) | 1.23 (0.72 to 2.09) | 1.14 (0.47 to 2.73) |
| Covert contraceptive use | **2.84 (1.21 to 6.64)*** | **2.99 (1.56 to 5.74)\*\*\*** | 0.83 (0.36 to 1.91) |
| | **Cote d'Ivoire** | | |
| | RC | Emotional IPV | Physical/sexual IPV |
| | aOR† (95% CI) | | |
| Current contraceptive use | 1.3 (0.72 to 2.34) | 1.15 (0.67 to 1.97) | 1.25 (0.61 to 2.56) |
| Method mix, among users | | | |
| Male compliant | ref | ref | ref |
| Female controlled | ‡ | 1.2 (0.67, 2.14) | ‡ |
| Covert contraceptive use | **4.45 (1.76 to 11.25)\*\*** | 1.88 (0.87 to 4.05) | 0.59 (0.19 to 1.86) |
| | **Kenya** | | |
| | RC | Emotional IPV | Physical/sexual IPV |
| | aOR† (95% CI) | | |
| Current contraceptive use | 0.84 (0.57 to 1.23) | **1.44 (1.04 to 2.01)*** | 0.82 (0.60 to 1.13) |
| Method mix, among users | | | |
| Male compliant | ref | ref | ref |
| Female controlled | 0.97 (0.55 to 1.69) | **0.73 (0.50 to 1.05)±** | 1.18 (0.67 to 2.07) |
| Covert contraceptive use | **5.77 (3.51 to 9.46)\*\*\*** | 0.8 (0.45 to 1.42) | **2.35 (1.33 to 4.17)\*\*** |

±<0.1, *<0.05, **<0.01, ***<0.001.
Bolding represents significance p<0.05.
Male compliant: male sterilisation, male condom, female condom, diaphragm, foam/jelly, standard days/cycle beads, rhythm, withdrawal, other traditional; female controlled: female sterilisation, implant, IUD, injectables, pills, EC, LAM.
†aOR: adjusted for other forms of violence, residence, education, parity, marital status, polygyny, employment.
‡Cells n<10 suppressed for regression.
aOR, adjusted odds ratio; EC, emergency contraception; IPV, intimate partner violence; IUD, intrauterine device; LAM, lactational amenorrhoea method; RC, reproductive coercion.

experiencing RC and covert use, as method switching may help minimise risk of partner suspicion.

Consistent with Demographic and Health Survey data, associations between physical/sexual IPV and contraceptive use were statistically insignificant,[17] however, emotional IPV was associated with significant increases in current contraceptive use in Kenya (aOR 1.44; 95% CI 1.04 to p<0.05) and trend towards association in Burkina Faso (aOR 1.36; 95% CI 0.97 to 1.92; p=0.06). These results may signify important benchmarks on the abuse cycle and pathways to help-seeking—women wishing to avert pregnancy within violent relationships are acting on their reproductive preferences to use contraception. Further, they justify the inclusion of emotional abuse measures within large-scale surveys given its high prevalence and concurrence with RC.

Results must be considered in light of limitations. Foremost, findings are cross-sectional, limiting conclusions surrounding temporality of associations—longitudinal work is needed to disentangle whether women are using contraception because of experienced violence or whether the violence is incurred given contraceptive use. Further, while RC measures have improved since this form of violence of first conceptualised, abortion coercion and forced contraceptive use are still not components of the RC scale, limiting our ability to capture a more inclusive range of RC experiences. Our RC measures also did not account for condom use manipulation and thus may undercount the true burden of RC in these contexts. Additionally, pregnant women were not included in the analytical sample, which could further underestimate RC prevalence as these women may be pregnant due to RC. However, including pregnant women in the sample would also lead to a systematic attenuation of the results between RC and contraceptive use given that these women are not in need of contraception. Women in dating

partnerships were also excluded from the present analysis due to embedded survey skip logic, though evidence among urban adolescents and young women indicates high prevalence of RC[29]; further research and oversampling is needed to understand reproductive health needs for this high risk subpopulation. Social desirability biases and privacy concerns could further contribute to under-reporting of experiences of violence, despite extensive training and privacy protocols aligned with best practices.[22 23]

Importantly, policy-makers and family planning and violence service providers must recognise RC and its health impact to holistically support women's needs. Work to dismantle deep-rooted social norms that dictate spousal permission prior to contraceptive provision must be included in healthcare provider trainings on how to support women experiencing RC. Interventions, such as Addressing Reproductive Coercion in Health Settings in Kenya,[30] and have proven useful in decreasing RC through training family planning service providers to recognise RC as a prevalent form of violence that impacts women's reproductive health[30]—scale-up of such programmes may help prevent recurrent violence. Integration of violence and reproductive health prevention and response services can further guarantee that women's needs are met regardless of type of abuse incurred. Standardisation of recommendations can further ensure that women are able to enact safety strategies to protect their reproductive health while concurrently mitigating the impact of abuse. Though many countries have made recent notable steps towards GBV prevention and response, national policies do not currently recognise RC. Policies must not only name RC as a detriment to women's health, but also include practical rights-based solutions that ensure women's privacy in contraceptive decision-making and freedom from partner interference, such as universal, affordable and judgement-free provision of covert contraceptive methods and EC, to ultimately counteract its reproductive health impact.

**Author affiliations**
[1]Department of Population, Family and Reproductive Health, Johns Hopkins University Bloomberg School of Public Health, Baltimore, Maryland, USA
[2]Bill & Melinda Gates Institute for Population and Reproductive Health, Johns Hopkins University Bloomberg School of Public Health, Baltimore, Maryland, USA
[3]International Centre for Reproductive Health Kenya, Mombasa, Kenya
[4]Institut Supérieur des Sciences de la Population, Ouagadougou, Centre, Burkina Faso
[5]Ecole Nationale Superieure de Statistique et d'Economie Appliquee, Abidjan, Côte d'Ivoire
[6]Technical University of Mombasa, Mombasa, Kenya
[7]Department of Public Health and Primary Care, Ghent University, Ghent, Belgium
[8]Johns Hopkins School of Nursing, Baltimore, Maryland, USA

**Contributors** SNW: study design, oversight, analysis, initial draft. HLT: analysis, graphics, initial draft. MT, GG, FB, YO, RM, RF, PG: study design, oversight of data collection, ethical adherence, interpretation and editing. MRD: study design, oversight, initial draft. All authors participated in writing and approving the final manuscript. SNW and MRD are the guarantors.

**Funding** This work was supported, in whole, by the Bill & Melinda Gates Foundation (009639). Under the grant conditions of the Foundation, a Creative

**Competing interests** None declared.

**Patient and public involvement** Patients and/or the public were involved in the design, or conduct, or reporting, or dissemination plans of this research. Refer to the Methods section for further details.

**Patient consent for publication** Not applicable.

**Ethics approval** All participants provided oral consent to participate. Approval was received by ethical review committees at Johns Hopkins School of Public Health, Comite D'Ethique Pour La Recherche en Sante, Ministere de la Recherche Scientifique et de L'Innovation, Ministere de la Sante in Burkina Faso, Comité National d'Ethique des Sciences de la Vie et de la Santé (CNESVS) in Côte d'Ivoire, and Kenyatta National Hospital-University of Nairobi Ethics and Research Committee College of Health Sciences in Kenya.

**Provenance and peer review** Not commissioned; externally peer reviewed.

**Data availability statement** Data are available on reasonable request. Data are available by reasonable request from pmadata.org.

**ORCID iD**
Shannon N Wood http://orcid.org/0000-0003-4389-3526

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
