## [Reviewer comments · BMJ Open]

ARTICLE DETAILS

TITLE (PROVISIONAL)	The intersection of reproductive coercion and intimate partner violence: cross-sectional influences on women's contraceptive use in Burkina Faso, Côte d'Ivoire, and Kenya
AUTHORS	Wood, Shannon; Thomas, Haley; Thiongo, Mary; Guiella, Georges; Bazié, Fiacre; Onadja, Yentéma; Mosso, Rosine; Fassassi, Raimi; Gichangi, Peter; Decker, Michele

VERSION 1 – REVIEW

REVIEWER	Sheeran, Nicola Griffith University, Applied Psychology
REVIEW RETURNED	16-Aug-2022

GENERAL COMMENTS	Thank you for the opportunity to review this paper. Work on reproductive coercion outside of the USA and Australia is vitally important and I feel that this paper makes an important contribution. Overall, the manuscript shares some important findings on a timely topic. My comments are fairly minor and the main issue is one that was identified by the authors and is something that can not be fixed at this stage. That is - the measurement of RC. Should the authors have the opportunity to influence future data collection, I do think that the RC measure needs to be amended as it is not accurately capturing the construct. There are also better measures of IPV. That said, the paper does have interesting findings around the association between covert contraception and pregnancy promoting reproductive coercion. In some ways, it is probably not a surprise that women experiencing coercion and abuse that promotes pregnancy conceal their contraception use. I think future research needs to unpack what women do when faced with contraception sabotage and pregnancy preventing reproductive coercion (including forced/coerced abortion). I think there is scope for further discussion of this limitation to the study in the paper. Some other points. Given the high associations between single/separated relationship status and RC it was unclear why only partnered women were asked to complete RC measures. Similarly pregnant women may have experienced RC and/or IPV in the past 12 months - and that could in fact be why they are now pregnant. Further justification for inclusion/exclusion is needed. A small thing as we know that language is important; the authors chose to define their sample as women in need of contraception. I really struggled with this label. Who says they are in need? I wonder if there are better labels for the group, though must confess every suggestion I had was quite wordy (i.e., women of
--

	childbearing age for whom contraception might be an option :) This is not a major issue for me, but if you can come up with a better label, I would encourage you to consider it. I was very confused about how covert use of contraception was assessed. The paper states "Direct covert use assessment was utilized" – what does this mean? I appreciate that you provide a reference but more information is needed in this paper for how that variable was created. I also found the explanation for the how the 3 categorical variables were created unclear. It did not become clear that it was female yes/no, male yes/no and covert yes/no until I read table 3. Currently the method is written as male-female and covert. I recommend making this a little clearer. Similarly, the contraceptive methods shown in the table 1 do not match the categories outlined in the method section. It is unclear why only a few of the specific forms of contraception were chosen for inclusion in the table and not others. Could you provide some definition of 'traditional' method as well. As a final note, I found the rates of pregnancy promoting RC interesting and similar to a paper we recently published. I also really enjoyed the Venn diagrams and found them thought provoking as we grapple with the intersections between RC and other forms of violence. I look forward to this paper being published.
--	---

REVIEWER	Hartman, Erin LSHTM, Public Health and Policy
REVIEW RETURNED	23-Oct-2022

GENERAL COMMENTS	Abstract: 25: list countries in parenthesis, smaller font is hard to read. 29: include numerical value of overlap between physical/sexual IPV and emotional IPV Key Messages: 6: suggest explaining how IPV and RC interact both concurrently and independently 10: change distinct contexts to "sampled contexts" 18: it would be important/interesting to comment on potential challenges (I.e. political) around promoting covert use Introduction: 5: change impact to affected or suffer and use a stronger word for hinder 12: delete trajectories 15: recommend remaining consistent in either calling it violence or abuse (and, I recommend calling it violence) 17-18: change to "common in sub-Saharan Africa" 18: change to "though behaviors AND intentions and motivations behind RC may differ" 22: provide examples of broader behaviors present in these contexts 25-26: reword this sentence, does not make much sense before the examples 36: may want to define emotional abuse, or at least give some examples 40: delete word outcomes 41: concurrently instead of simultaneously
---

	Methods: 3: explain why these three countries were selected 29: what does emotional upset mean? 52: considering acknowledging, as a limitation, that only currently married/partnered were included (page 7) 18: why only the pregnancy sub-scale (and maybe explain why so) Results: 7: comment on the percentage neither married nor partnered? (page 10) 7: suggest different word for mistreatment - or just explain what this includes (page 11): explain what methods were considered covert - and who labeled them this? the women themselves? Discussion: 18: need more explanation of what were covert/female controlled methods (page 15): add considerations of promoting covert use
--	---

VERSION 1 – AUTHOR RESPONSE

Response to Reviewer 1:

1. Thank you for the opportunity to review this paper. Work on reproductive coercion outside of the USA and Australia is vitally important and I feel that this paper makes an important contribution.

Authors' Response: Thank you for your kind and thoughtful review of our manuscript.

2. Overall, the manuscript shares some important findings on a timely topic. My comments are fairly minor and the main issue is one that was identified by the authors and is something that can not be fixed at this stage. That is - the measurement of RC. Should the authors have the opportunity to influence future data collection, I do think that the RC measure needs to be amended as it is not accurately capturing the construct. There are also better measures of IPV. That said, the paper does have interesting findings around the association between covert contraception and **pregnancy promoting reproductive coercion. In some ways, it is probably not a surprise that women experiencing coercion and abuse that promotes pregnancy conceal their contraception use. I think future research needs to unpack what women do when faced with contraception sabotage and pregnancy preventing reproductive coercion (including forced/coerced abortion). I think there is scope for further discussion of this limitation to the study in the paper.**

Authors' Response: Thank you for this suggestion--we agree that these are limitations and that future research should be amended to more thoroughly address this construct. We agree that the limitations of the RC scale are important to acknowledge and have expanded the scope of the discussion to include these limitations (pg. 13, line 39-43).

3. Given the high associations between single/separated relationship status and RC it was unclear why only partnered women were asked to complete RC measures. Similarly pregnant women may have experienced RC and/or IPV in the past 12 months - and that could in fact be why they are now pregnant. Further justification for inclusion/exclusion is needed.

Authors' Response: We clarify that only women who were married or living with a partner as if married were asked the RC questions given the item wording necessitating partnership (i.e. in

the past 12 months, has your husband or partner..”). We have clarified this wording in text (pg. 6, line 13; pg. 6, line 15; pg. 6, line 20). We recognize that our exclusion of pregnant women may lead to an underestimate of RC, however, it would also lead to a systematic attenuation of the results between RC and contraceptive use given that these women are not in need of contraception. We have added language to recognize this restriction as a limitation (pg. 13, line 45-pg. 14, line 1).

4. A small thing as we know that language is important; the authors chose to define their sample as women in need of contraception. I really struggled with this label. Who says they are in need? I wonder if there are better labels for the group, though must confess every suggestion I had was quite wordy (i.e., women of childbearing age for whom contraception might be an option :) This is not a major issue for me, but if you can come up with a better label, I would encourage you to consider it.

Authors’ Response: Thank you for this comment. We clarify that this definition is consistent with the family planning indicator for unmet need for contraception. In order to ensure that these results are applicable to family planning and violence stakeholders, practitioners, and researchers, we would like to keep the standard definition wording.

5. I was very confused about how covert use of contraception was assessed. The paper states "Direct covert use assessment was utilized" – what does this mean? I appreciate that you provide a reference but more information is needed in this paper for how that variable was created. I also found the explanation for the how the 3 categorical variables were created unclear. It did not become clear that it was female yes/no, male yes/no and covert yes/no until I read table 3. Currently the method is written as male-female and covert. I recommend making this a little clearer. Similarly, the contraceptive methods shown in the table 1 do not match the categories outlined in the method section. It is unclear why only a few of the specific forms of contraception were chosen for inclusion in the table and not others. Could you provide some definition of 'traditional' method as well.

Authors’ Response: Many thanks for these questions. You are correct in reading this as male-female and covert yes-no as two separate outcomes. We have added further information on the three dichotomous variables (pg. 6, lines 31-38). We also clarify that covert contraceptive use has traditionally been asked in large-scale surveys, such as the Demographic and Health Survey, in two ways: 1) by directly asking women if they are using contraception without their partner’s knowledge (direct question) or 2) by asking both men and women if the woman is using contraception and then comparing reports to see if the partner knows of use (indirect question). We have included the survey item that was used to assess covert contraceptive use for clarity (pg. 6, lines 39-42).

We have further updated the method mix for Table 1 to increase clarity and ensure consistency with the method mix categories for further analyses. These are now split into male compliant (male sterilization, male and female condoms, diaphragm, foam/jelly, standard days/cycle beads, rhythm, withdrawal, other traditional), female-controlled LARC (female sterilization, implant, IUD), and female-controlled short-acting (injectables, pills, EC, LAM) (Table 1).

6. As a final note, I found the rates of pregnancy promoting RC interesting and similar to a paper we recently published. I also really enjoyed the Venn diagrams and found them thought provoking as we grapple with the intersections between RC and other forms of violence. I look forward to this paper being published.

Authors’ Response: Many thanks again for your kind words and thoughtful review.

Response to Reviewer 2

Abstract:

1. 25: list countries in parenthesis, smaller font is hard to read.

Authors' Response: We have updated the format to remove the smaller font (Abstract).

2. 29: include numerical value of overlap between physical/sexual IPV and emotional IPV

Authors' Response: Thank you for this suggestion. Given the focus of this paper is largely on RC, we do not report these values in text and therefore feel that reporting them within the Abstract would detract from the main results.

Key Messages:

3. 6: suggest explaining how IPV and RC interact both concurrently and independently
10: change distinct contexts to "sampled contexts"
18: it would be important/interesting to comment on potential challenges (I.e. political) around promoting covert use

Authors' Response: Per request of the editor, the Key Messages section has been removed altogether and replaced with a Strengths and Limitations section.

Introduction:

4. 5: change impact to affected or suffer and use a stronger word for hinder

Authors' Response: Thank you for these suggestions—we have edited the sentences accordingly (pg. 4, line 3).

5. 12: delete trajectories

Authors' Response: We would like to retain "trajectories" as this sentence is stating that RC impacts both women's current reproductive health and future reproductive health (pg. 4, line 8).

6. 15: recommend remaining consistent in either calling it violence or abuse (and, I recommend calling it violence)

Authors' Response: Thank you for your suggestion. We have updated the language to "violence" throughout for consistency.

7. 17-18: change to "common in sub-Saharan Africa"

Authors' Response: We have made the suggested edit (pg. 4, lines 13-14).

8. 18: change to "though behaviors AND intentions and motivations behind RC may differ"

Authors' Response: We are referring to behaviors, intentions, and motivations as three separate things, and would therefore like to keep the punctuation as is.

9. 22: provide examples of broader behaviors present in these contexts

Authors' Response: Thank you for this suggestion. We have included some examples of broader behaviors (pg. 4, lines 17-18).

10. 25-26: reword this sentence, does not make much sense before the examples

Authors' Response: We have updated this sentence to increase clarity (pg. 4, lines 21-23).

11. 36: may want to define emotional abuse, or at least give some examples

Authors' Response: Thank you for this suggestion. We have provided some examples of emotional abuse (pg. 4, lines 27-28).

12. 40: delete word outcomes

Authors' Response: Since we are examining three different contraceptive use outcomes—current contraceptive use, contraceptive method mix (female-controlled vs. male compliant), and covert use—we believe we should keep the wording as is.

13. 41: concurrently instead of simultaneously

Authors' Response: We have updated the wording of this sentence (pg. 4, line 36).

Methods

14. 3: explain why these three countries were selected

Authors' Response: Thank you for this suggestion. We have noted why these three countries were included in the analysis (pg. 5, lines 13-14).

15. 29: what does emotional upset mean?

Authors' Response: We have clarified the meaning of emotional upset by providing examples (pg. 5, lines 25-26).

16. 52: considering acknowledging, as a limitation, that only currently married/partnered were included

Authors' Response: Many thanks for this suggestion. We have added additional details to the discussion to acknowledge that our sample excluded those in dating partnerships, though these women could still experience RC (pg.14, lines 2-5).

17. (page 7) 18: why only the pregnancy sub-scale (and maybe explain why so)

Authors' Response: Thank you for this suggestion. We have clarified that we only use the pregnancy coercion sub-scale given low condom use in ongoing relationships and connotations surrounding condom use for disease prevention rather than pregnancy prevention in sub-Saharan Africa (pg.6. lines 24-26).

Results:

18. 7: comment on the percentage neither married nor partnered?

Authors' Response: We clarify that these women were not part of our sample. RC and IPV items were only asked to women who were married or living with a partner as if married; therefore, percentages will not be included.

19. (page 10) 7: suggest different word for mistreatment - or just explain what this includes

Authors' Response: This is the wording of an item in the RC Scale so we would like to retain it for consistency.

20. (page 11): explain what methods were considered covert - and who labeled them this? the women themselves?

Authors' Response: We did not classify specific methods as covert. Covert use was determined from a survey question—asked among women currently using a contraceptive method—if their partner knew they were using that method. Item wording has been clarified in the methods section (pg. 6, lines 41-42).

Discussion:

21. 18: need more explanation of what were covert/female controlled methods

Authors' Response: Additional information has been added to the Methods section for clarity on these variables (pg. 6, lines 30-42).

22. (page 15): add considerations of promoting covert use

Authors' Response: Thank you for this suggestion. We have included implications of promoting covert use in our discussion (pg. 13, lines 24-27).

VERSION 2 – REVIEW

REVIEWER	Sheeran, Nicola Griffith University, Applied Psychology
REVIEW RETURNED	07-Dec-2022

GENERAL COMMENTS	Thank you to the authors for their careful consideration of the suggestions for improving this paper. I feel that the revised paper is much clearer and really highlights the central important message pertaining to RC, IPV and contraception. This paper is timely and important and makes an excellent contribution to our understanding of RC and contraception use. One minor comment to correct when next proofing the paper - and that is to ensure that all acronyms are defined at time of first use (i.e., EC and LAM).
--

REVIEWER	Hartman, Erin LSHTM, Public Health and Policy
REVIEW RETURNED	27-Dec-2022

GENERAL COMMENTS	Thank you for making these revisions, they make the manuscript clearer. Thank you for this important piece, it makes a valuable contribution to the field and is helpful for your colleagues in this field. I have a few additional comments for your consideration:  1. p. 23 line 4: It is slightly confusing to describe it as an intersection, but to then list three items. I suggest making it clear that the intersection between physical and sexual IPV and RC will be investigated AND that the intersection between emotional IPV and RC will be investigated. 2. Why are the strengths and limitations now presented in the beginning of the article? 3. p. 28 line 27: I know this was using already-collected survey data, but it would be helpful to expand a bit more on why condoms were not included - given that they are more likely to be used for HIV/AIDS prevention, but they are not only used for this purpose. Perhaps, this is merely a limitation of the survey. 4. Would it be useful to briefly comment on the limitations and criticisms of using both the conflict tactics scale and the RCS? 5. p. 28: It may be helpful to elaborate on how contraceptive questions were framed - to just provide an example of a few of the questions asked. 6. Perhaps there is no room to elaborate, but rhythm and cycle beads being considered male-compliant methods seems nuanced and could use a bit more discussion.
--

	7. p. 29 line 4: Is this the only way to measure if use is covert? I am just thinking if there are other reasons why a partner wouldn't know if she was using FP - what if he doesn't care? For future studies, could consider asking first "does your partner know that you are using FP?" and then ask "if so, does your partner know that you're using X method?" 8. p. 35 line 27: Consider adding to side effects how a method impacts the partner (and then providing examples of such) as it seems like it's a combination of how women experience side effects and how these side effects then affect their partner. Thank you for the opportunity to review this manuscript. I wish you the best of luck finishing this process, and I am looking forward to sharing this publication and your findings with colleagues
--	---

VERSION 2 – AUTHOR RESPONSE

Many thanks to the Editor and Reviewers for their continued helpful feedback. Please see point-by-point response below.

Reviewer 1: Dr. Nicola Sheeran, Griffith University

Comments to the Author:

Thank you to the authors for their careful consideration of the suggestions for improving this paper. I feel that the revised paper is much clearer and really highlights the central important message pertaining to RC, IPV and contraception. This paper is timely and important and makes an excellent contribution to our understanding of RC and contraception use.

One minor comment to correct when next proofing the paper - and that is to ensure that all acronyms are defined at time of first use (i.e., EC and LAM).

Authors' Response: Thank you for your immensely helpful edits. We have proofread the paper to ensure that all acronyms are defined at first use.

Reviewer 2: Ms. Erin Hartman, LSHTM

Comments to the Author:

Thank you for making these revisions, they make the manuscript clearer. Thank you for this important piece, it makes a valuable contribution to the field and is helpful for your colleagues in this field.

Authors' Response: Many thanks for your continued review of our manuscript. We note that our version of page numbers was different than this reviewer's and therefore we have attempted to answer these points as best as possible.

I have a few additional comments for your consideration:

- 1. p. 23 line 4: It is slightly confusing to describe it as an intersection, but to then list three items. I suggest making it clear that the intersection between physical and sexual IPV and RC will be investigated AND that the intersection between emotional IPV and RC will be investigated.**

Authors' Response: Thank you for this point. While the primary focus of this paper is on RC's intersection with physical/sexual IPV and emotional IPV, the intersection between physical/sexual IPV and emotional IPV is depicted in the Venn diagrams. Therefore, we would like to retain this wording in the objectives.

2. Why are the strengths and limitations now presented in the beginning of the article?

Authors' Response: The strengths and limitations immediately following the abstract conform to BMJ Open submission guidelines and were requested by the Editors in the last review: https://bmjopen.bmj.com/pages/authors/#submission_guidelines.

3. p. 28 line 27: I know this was using already-collected survey data, but it would be helpful to expand a bit more on why condoms were not included - given that they are more likely to be used for HIV/AIDS prevention, but they are not only used for this purpose. Perhaps, this is merely a limitation of the survey.

Authors' Response: We have included this point in the manuscript "The condom-manipulation subscale of the RC Scale was not included given that condoms are more likely to be used in these settings for preventing HIV/STIs than pregnancy" (pg. 6, lines 24-26; pg. 13, lines 45-46). We further clarify that condoms are more likely to be used within transient partnerships, rather than those ongoing, and the study sample is limited to those married or cohabitating as if married. Additional research with unmarried and young women is needed to better understand condom manipulation measures.

4. Would it be useful to briefly comment on the limitations and criticisms of using both the conflict tactics scale and the RCS?

Authors' Response: We are unclear on this Reviewer's point. We clarify that we have included criticisms of the RCS on pg. 13, lines 42-46: "Further, while RC measures have improved since this form of violence of first conceptualized, abortion coercion and forced contraceptive use are still not components of the RC scale, limiting our ability to capture a more inclusive range of RC experiences. Our RC measures also did not account for condom use manipulation and thus may undercount the true burden of RC in these contexts." Both the RCS and CTS-2 are behavioral assessments in line with best practices on violence-related research.

5. p. 28: It may be helpful to elaborate on how contraceptive questions were framed - to just provide an example of a few of the questions asked.

Authors' Response: Given that these are standard assessments and fully available online, we have opted to not include the framing. We further clarify that PMA surveys are focused on contraceptive use, thus, there was no segue into these questions.

6. Perhaps there is no room to elaborate, but rhythm and cycle beads being considered male-compliant methods seems nuanced and could use a bit more discussion.

Authors' Response: We have added a line to clarify why cycle tracking methods are included in this category (pg. 6, line 39).

7. **p. 29 line 4: Is this the only way to measure if use is covert? I am just thinking if there are other reasons why a partner wouldn't know if she was using FP - what if he doesn't care? For future studies, could consider asking first "does your partner know that you are using FP?" and then ask "if so, does your partner know that you're using X method?"**

Authors' Response: We clarify that the covert use item is asked in line with the previous wording for the direct covert use measure within the Demographic and Health Surveys. We further clarify that this item is only assessing whether the partner knows of use (i.e., covert), and not partner support and/or involvement in contraceptive use. Lastly, this is a secondary data analysis of a large-scale survey, and thus, these items are not modifiable.

8. **p. 35 line 27: Consider adding to side effects how a method impacts the partner (and then providing examples of such) as it seems like it's a combination of how women experience side effects and how these side effects then affect their partner.**

Authors' Response: This suggestion seems to be about adapting future survey items, rather than about the measures utilized for the current analysis. We again emphasize that this is a secondary data analysis, and the questionnaire was not modifiable.

Thank you for the opportunity to review this manuscript. I wish you the best of luck finishing this process, and I am looking forward to sharing this publication and your findings with colleagues.

Authors' Response: Many thanks for your continued review and contribution to this field.